# HYBRIDCOT: INTERLEAVING LATENT AND TEXT CHAIN-OF-THOUGHT FOR EFFICIENT REASONING

## ABSTRACT

Verbalizing intermediate steps in token space has been central to eliciting reasoning in large language models (LLMs), with longer reasoning generally improving performance but incurring substantial compute and memory costs. Prior attempts to improve efficiency—such as KV-pruning or latent-space reasoning—often suffer from loss of accuracy or training inefficiency. We propose HybridCoT, a framework that interleaves latent and text reasoning tokens in context. Our method reduces the compression errors that troubles previous latent CoT methods by keeping critical text tokens like math operations, in context, while compress semantic reasoning into the latent space. In addition, we design in-context text-to-token distillation to provide explicit supervision and iterative parallelized latent rollout methods to improve training efficiency for latent token, while shortening reasoning paths for efficiency. On challenging math reasoning benchmarks including AIME and MATH, HybridCoT achieves 94% of the performance of finetuned text-only CoT models with 1.97× less inference compute, and surpasses efficient baselines (LightThinker and StreamLLM) by 1.36× and 1.26×, respectively.

## 1 INTRODUCTION

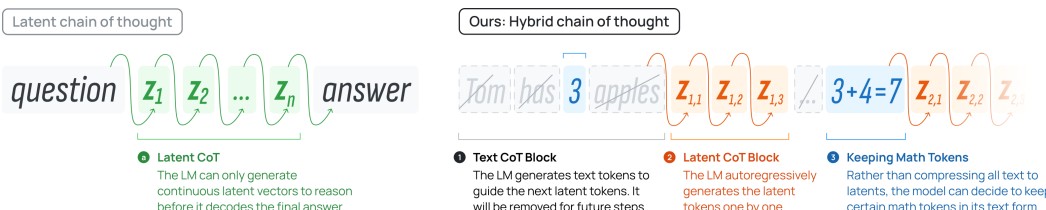

Fig 1: We train a language model to alternate between text and latent reasoning modes within the same reasoning trace. It outputs text tokens first, which support the generation of succeeding latent tokens. The text tokens are then removed from future context to speed up inference, while it retains math tokens in the context, which turn out to be crucial for math reasoning tasks.

Large language models (LLMs) have achieved remarkable success on complex reasoning tasks by generating extended reasoning sequences, known as chain-of-thought (CoT) reasoning (Wei et al., 2022; Muennighoff et al., 2025; OpenAI, 2024). Beyond reasoning verbally in text space, there is growing interest in latent reasoning using dense vector representations, or latent tokens (Hao et al., 2024; Cheng & Durme, 2024; Shen et al., 2025). This approach offers several compelling advantages: latent tokens can compress multiple reasoning steps into fewer representations, reducing context length and memory requirements; they enable more flexible computation allocation rather than uniform token budgets; and they can potentially capture reasoning patterns that are difficult to express in natural language Zhu et al. (2025); Chen et al. (2025).

However, training models to reason with long latent CoT traces comes with significant challenges. **First, compression errors arise** as existing latent CoT methods require a homogeneous context of latent tokens. Storing all information and reasoning in latent space can introduce unintended errors, especially for symbolic tokens like numbers and mathematical operators, as found in previous literature (Zhang et al., 2025). **Second, latent tokens lack direct supervision signals** since

training data contains only textual reasoning traces. Previous methods leverage indirect supervision through curriculum learning with text removal (Hao et al., 2024) or by distilling certain latents from a teacher model (Cheng & Durme, 2024; Shen et al., 2025), but such indirect supervision becomes less effective as reasoning length increases. **Third, latent CoT model training is computationally expensive**, often requiring multiple times more time compared to text CoT training even for shorter reasoning traces (Hao et al., 2024; Shen et al., 2025), and this overhead becomes worse as reasoning traces grow longer (as it needs to multiple forward passes for generating all latent token before a gradient update). As such, current latent CoT methods only demonstrate success in relatively simple tasks like GSM8K (Cobbe et al., 2021) with short reasoning.

We propose Hybrid-CoT, a method to combine text and latent CoT within the same reasoning trace. As illustrated in Fig. 1, the model is trained to **alternate between text and latent reasoning modes** for a given problem. Upon generating a few text tokens, the model can switch to decoding a fixed number of latent tokens. The text tokens are primarily used to support the generation of the succeeding latent tokens and will be removed from the future context to speed up inference. This compression-like objective (Mu et al., 2023; Zhang et al., 2025) can provide fine-grained supervision signals, and we find that the trained models can produce longer latent sequences of up to a few thousand tokens.

Notably, instead of removing all text tokens from the preceding context, we also train the model to selectively keep certain tokens resulting in a **hybrid context with both text and latent tokens**. We find that retaining math tokens in the context has a significant impact on the downstream performance of the models: it avoids possible information loss during the compression or reconstruction process, and the LM can leverage existing induction heads (Olsson et al., 2022) for symbolic processing of the text tokens.

To make training of long latent CoT possible, we propose an efficient algorithm to approximate latent tokens with constant complexity irrespective of reasoning length, compared to the linear complexity of existing methods (Hao et al., 2024). Since we are targeting very challenging math problems like those in AIME (MAA, 2024), existing methods would slow down training by hundreds of times—as the model often needs to reason for hundreds of steps.

We demonstrate that models trained with Hybrid-CoT can achieve strong performance on math and general reasoning benchmarks including AIME24-25 (MAA, 2024; 2025), AMC (MAA, 2023), MATH (Hendrycks et al., 2021), and GPQA (Rein et al., 2023)—HybridCot matches 94% performance of text CoT across all tasks (long-CoT benchmarks plus MATH and GPQA) with 1.97x less compute; it outperforms efficient baselines LightThinker and StreamLLM by 1.36x and 1.26x on average with Qwen3-8B and Qwen2.5-7B, respectively. In addition, HybridCoT demonstrate significant improvement over other efficient baselines on complex math benchmarks that require long CoT such as AIME (MAA, 2024) and AMC (MAA, 2023)—Hybrid-CoT achieves an averaged score of 66.53% across these long-CoT benchmarks, in comparison with 37.57% for LightThinker (Zhang et al., 2025) and 28.75% for StreamLLM (Xiao et al., 2023) with Qwen3-8B. We will fully release our code to facilitate the research community to build on top of our method.

## 2 PRELIMINARIES AND RELATED WORKS

Latent chain-of-thought (CoT) and context compression represent two prominent approaches for reducing computational complexity in Transformer-based LLMs. Latent CoT generates information-rich dense vectors in continuous space, while context compression directly reduces the token count in the input context. In this section, we present a unified view on these research streams and identifies the existing challenges that motivate our study.

**Language model and text CoT.** For an input sequence of $n$ tokens $X = [x_1, x_2, \ldots, x_n] \in \mathcal{V}^n$ from a discrete vocabulary $\mathcal{V} \subset \mathbb{Z}$, an autoregressive LM $\mathcal{F}$ outputs a categorical distribution $\mathcal{F}(\cdot | X_{<i})$ over $\mathcal{V}$ for the $i$-th token given the prefix $X_{<i}$. In transformer-based (Vaswani et al., 2017) LMs, each token $x_i$ is embedded into a $d$-dimensional space $\mathbf{e}_i \in \mathbb{R}^d$. They are processed through a stack of model layers, where each layer mixes the embeddings along both the $n$ and $d$ dimensions. The final layer outputs a continuous vector $\mathbf{z}_i \in \mathbb{R}^d$ for the $i$-th token, which is then mapped to the logits over $\mathcal{V}$ using a linear projection. Notably, Transformers require quadratic computational complexity with respect to context length, making long chain-of-thought reasoning

computationally expensive. By autoregressively decoding the next tokens, the LM can generate a sequence of length $n$ representing the reasoning process for a given user problem $\mathbf{q}$, and recent work (Muennighoff et al., 2025; OpenAI, 2024) shows that as $n$ increases, the model can solve increasingly difficult tasks, a phenomenon often dubbed "inference time compute scaling". However, one common critique of text CoT is that it assigns uniform compute per decoding step, and it calls for a more flexible allocation of compute for different steps given the difficulty of the reasoning.

**Compressing text CoT with sparse attention.** As the quadratic computational cost introduced by increasing KV-cache size becomes the primary bottleneck for long-form reasoning (Austin et al., 2025), context compression approaches improve inference efficiency by *reducing the prefix size* per decoding step. One way is to have the LLM generate full text CoT and retain only a subset of tokens in the context for each step. The subsetting can be a static sliding window (Xiao et al., 2023) or dynamically determined (e.g., evicting certain tokens from the KV-cache (Zhang et al., 2023)). This approach does not create explicit latent representations but instead relies on the language model to implicitly learn to utilize the remaining contextual information effectively: as such, there's substantial performance drop above certain compression ratios(Zhang et al., 2023).

**Reasoning with latent CoTs.** Recent works have explored using the hidden states of the LLM as latent CoTs for the intermediate reasoning steps. Specifically, this approach involves **autoregressively decoding using dense vectors** $\mathbf{z}_i$ that forms a "latent chain of thought" (Hao et al., 2024, latent CoT). It needs *fewer decoding steps* given that latent vectors can be more flexible and carry additional information (Zhu et al., 2025), thus improving the inference speed. However, there are two challenges during training: **it lacks direct supervision for latent COTs**, as training data only contains textual CoTs; and **the training is very inefficient** as it requires autoregressively decoding $\mathbf{z}_i$ to obtain policy latents before each backward pass (whereas standard training only requires a single pass of forward per training step), which can slow down training by $n$-fold, where $n$ is the number latent tokens. Despite recent attempts to address some of these challenges (Shen et al., 2025; Cheng & Durme, 2024; Wang et al., 2025), latent CoT methods are still limited to relatively simple, short-form reasoning tasks, tailing behind the performance of text CoT methods.

**Gisting.** Recent approaches also explore interleaving text generation with "soft tokens" that compress the context. For example, context gisting (Mu et al., 2023) trains an LM to store relevant information of the prompt in the activations values for fixed gist tokens $[g_1, \ldots, g_m], g \in \mathcal{V}_{\text{gist}}$. LightThinker (Zhang et al., 2025) extends this method to iteratively compress the reasoning with the same set of gist tokens for multiple reasoning steps. On the other hand, Chevalier et al. (2023) uses the language model to recursively generate an $m$-sized block of summary vectors $[\mathbf{s}_{(b,1)}, \ldots, \mathbf{s}_{(b,m)}], \mathbf{s}_{(b,i)} \in \mathbb{R}^d$ for the $b$-th chunk in the context, using the transformer LM itself. Noteworthily, upon generating the soft tokens, the methods **remove all preceding text tokens**, leading to potential information loss as reported by Zhang et al. (2025). By interleaving the text generation and compression steps, the model can have more fine-grained supervision signal for the soft tokens, and they can scale to longer reasoning tasks. However, given the complexities of latent rollout as mentioned above, there is **no or limited recurrence among soft tokens**. While they may involve continuous vectors in the context, their primary focus is to compress rather than to advance the reasoning process of each step.

## 3 METHOD

### 3.1 HYBRID TEXT AND LATENT CHAIN OF THOUGHT

**Interleaving text and latent reasoning blocks.** Recall that we define a *reasoning block* as a subsequence of the full reasoning trace, segmented by sentences or paragraphs. We train the LM to alternate between text and latent reasoning modes in discrete blocks. Formally, we define $X[b] = [x_{(b,1)}, x_{(b,2)}, \ldots, x_{(b,k_b)}]$ as the $b$-th block of text tokens of length $k_b$, and $Z[b] = [\mathbf{z}_{(b,1)}, \mathbf{z}_{(b,2)}, \ldots, \mathbf{z}_{(b,m)}]$ as the corresponding block of $m$ latent vectors $\mathbf{z} \in \mathbb{R}^d$. We note that $k_b$ varies for different blocks, while $m$ is fixed as a hyper-parameter—in each reasoning block, the model first generates $k_b$ text tokens $x_{(b,i)} \in \mathcal{V}$ from the vocabulary for a draft of textual reasoning; Upon generating the special token `<latent>`, which indicates the completion of textual reasoning for that block, the model switches to latent reasoning mode and autoregressively decodes a

fixed number of $m$ latent tokens. Compared with previous method (Zhang et al., 2025) that uses a fixed set of special text tokens for compressing all reasoning blocks, our approach uses flexible, continuously-valued latent vectors to carry more information through autoregressive generation and advance the reasoning process.

**Mixing text and latent tokens in the context.** Before moving to the next block, our method can opt to retain a subset of text tokens in the $b$-th stage. In contrast to typical context compression methods that discard all text tokens from previous blocks and rely solely on compressed representations, our approach selectively preserves important textual information. Formally, we define $X'[b] = [x_{(b,i)}]_{i \in S_b}$ as the selected text tokens from block $b$, where $S_b \subseteq \{1, 2, \ldots, k_b\}$ is the set of indices for tokens to be retained. It is inspired by the observation that certain parts of the reasoning process require a more precise, exact mode (e.g., math computation), and fully relying on latent vectors for such reasoning may cause unintended errors (Zhang et al., 2025). While one could train models to automatically learn optimal text selection policies (Shen et al., 2024; Akhauri et al., 2025), we adopt a simple approach for our mathematical reasoning experiments: we construct the index set $S_b$ for tokens of math symbols, numbers, and computations.[1] In practice, we prompt a powerful language model to label math text spans with `<math>` in the training data (see Section A for details). During inference, the LLM generates `<math>` tags, the tokens between which will be preserved in its original textual format, as illustrated below.

*Text CoT* $X[1]$ $[x_{(1,1)}, \ldots, x_{(1,k_1)}]$

*Latent CoT* $Z[1]$ $[x_{(1,1)}, \ldots, x_{(1,k_1)}, \mathbf{z}_{(1,1)}, \ldots, \mathbf{z}_{(1,m)}]$

*Select Text* $X'[1]$ $[x_{(1,1)}, \ldots, x_{(1,j_1)}, \ldots, x_{(1,j_{l_1})}, \ldots, x_{(1,k_1)}, \mathbf{z}_{(1,1)}, \ldots, \mathbf{z}_{(1,m)}], \; j_1, \ldots, j_{l_1} \in S_1$

To formalize the per-token generation process, we use the notation $X[b,i] = [x_{(b,1)}, \ldots, x_{(b,i)}]$ to denote the first $i$ elements of sequence $X[b]$, and similarly for $Z[b,i]$. We define $\mathcal{C}[b,i]$ as the hybrid context till the $i$-th latent token in the $b$-th block:

$$\mathcal{C}[b,i] = X'[1] \oplus Z[1] \oplus \cdots \oplus X'[b] \oplus Z[b,i], \tag{1}$$

where $\oplus$ denotes list concatenation.

In the text CoT stage, the LM generates the next token following a categorical distribution over the vocabulary: $x_{(b,i)} \sim \mathcal{F}(\cdot | \mathcal{C}[b-1, m] \oplus X[b, i-1])$ for the $i$-th text token in the $b$-th block; in the latent CoT stage, it outputs the latent token $\mathbf{z}_{(b,i)} = \mathcal{F}_{\text{latent}}(\mathcal{C}[b, i-1])$ given the hybrid context. Different from COCONUT (Hao et al., 2024) that directly uses the LM's last hidden state as the latent token, we apply a linear layer to project the last hidden state to the latent vectors, which is trained jointly with the rest of the model.

**Training objective.** Our method modifies the standard instruction fine-tuning pipeline on reasoning data (Muennighoff et al., 2025) to support hybrid reasoning. For each textual reasoning trace in the training data, we first preprocess the data to split it into multiple blocks of reasoning by sentences or by paragraphs and insert the `<latent>` tags. During training, the model generates latent tokens after each text reasoning block and learns to compress the text reasoning into latent tokens. During inference, the model can automatically generate

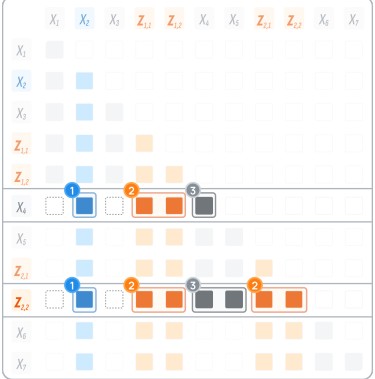

Fig 2: We add sparsity in the attention matrix: for each token, it can only attend to *(1) math tokens kept from previous blocks*, *(2) all previous latent tokens* and *(3) all text tokens in the current block*.

text tokens wrapped with `<latent>` tags and following latent tokens. Once it finishes the latent token generation, the preceding text tokens will be removed from the context except those in `<math>` tags. The hybrid context is implemented using a sparse attention mask. As illustrated in Fig. 2, within a reasoning block containing both text and latent tokens, the sparse attention matrix ensures that each token attends only to its designated context, effectively mimicking token removal.

---

[1]This strategy serves as a proof-of-concept, and the framework can be generalized to other domains by developing appropriate token selection criteria accordingly.

Following Hao et al. (2024), we compute the cross-entropy loss on all the text tokens in the context. Since we interleave text and latent tokens, the loss on the text token can provide fine-grained supervision for each latent reasoning stage.[2] Unlike prior work, our design retains text tokens in the context when training latent tokens within the same reasoning block, providing direct supervision that can help stabilize and accelerate training. At the same time, these text tokens are masked out from the context of inter-block latent tokens, ensuring that no information leaks across blocks.

## 3.2 EFFICIENT TRAINING WITH ITERATIVE PARALLELIZED LATENT ROLLOUT

Compared to text CoT, one important difference in latent CoT training is that it requires the model to autoregressively generate, or "roll out", the latent tokens before *every* loss computation and gradient update. In existing methods (Hao et al., 2024; Shen et al., 2025), it requires $B \times m$ forward passes to generate all latent tokens for a reasoning trace consisting of $B$ blocks with $m$ tokens each.[3] As a result, the complexity becomes a multiple of the number of latent tokens in all reasoning blocks $B \times m$, compared to standard text-only training, making it computationally expensive to train on long reasoning tasks (as it slows down the training by $\mathcal{O}(B)$ times).

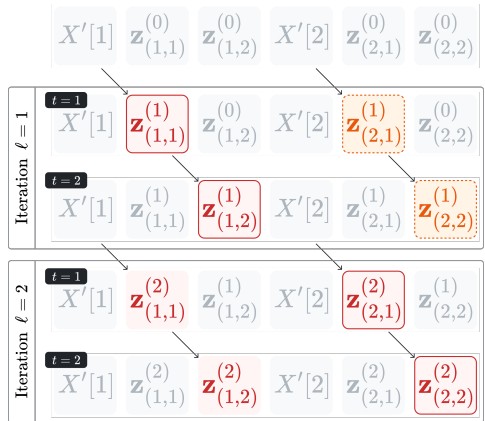

Fig 3: In iterative latent rollout, after the $\ell$-th full iteration, the first $\ell$ blocks of latent tokens become exact (colored red) and stop updating (borderless).

To address this issue, we present an iterative algorithm that can approximate latent tokens with only $L \times m$ forward passes, where $L$ is a small constant hyper-parameter and $L \ll B$. The intuition behind this method is to relax the causal dependencies across reasoning blocks, which enables greater parallelism during forward computation. At the same time, we apply iterative updates to progressively reduce approximation error. This design increases the number of forward passes only by a much smaller factor, rather than scaling with the number of latent tokens, thereby greatly reducing computational cost.

Formally, we denote $\tilde{\mathbf{z}}_{(b,i)}^{(\ell)}$ as the value of the latent token $\mathbf{z}_{(b,i)}$ in the $\ell$-th iteration, and we set the initial value of the $i$-th latent token in *all* blocks as $\tilde{\mathbf{z}}_{(b,i)}^{(0)} = \mathbf{v}_i, \forall b \in [1, B]$, which is initialized as the token embeddings of textual gist tokens $[g_1, \ldots, g_m], g \in \mathcal{V}_{\text{gist}}$ introduced in Section 2. At the $t$-th token generation in the $\ell$-th iteration, the latent tokens in the $b$-th block is

$$\tilde{Z}_t^{(\ell)}[b] = [\quad \overbrace{\tilde{\mathbf{z}}_{(b,1)}^{(\ell)}, \quad \ldots, \quad \tilde{\mathbf{z}}_{(b,t)}^{(\ell)}}^{\text{Newly updated tokens in the } \ell\text{-th iteration}}, \quad \overbrace{\tilde{\mathbf{z}}_{(b,t+1)}^{(\ell-1)}, \quad \ldots, \quad \tilde{\mathbf{z}}_{(b,m)}^{(\ell-1)}}^{\text{Rest tokens from the previous iteration}} ]. \quad (2)$$

Next, we use $\mathcal{C}_t^{(\ell)}[b, i]$ to denote the updated hybrid context with the latent tokens updated to $\tilde{Z}_t^{(\ell)}[b]$.

As shown in Fig. 3, inside the $\ell$-th iteration, when decoding the $t$-th token, the LM can compute the $t$-th latent tokens for all blocks in parallel: $\tilde{\mathbf{z}}_{(b,t)}^{(\ell)} = \mathcal{F}_{\text{latent}}(\mathcal{C}_{t-1}^{(\ell)}[b, t-1]), \forall b \in [1, B]$. We update the $t$-th latent tokens in all blocks, which will be used to approximate the subsequent latent tokens. After generating $m$ latent tokens for all blocks in parallel, one iteration is completed. The generated latent tokens are then used as inputs for the next iteration if $\ell < L$; otherwise, the process terminates and the final latent tokens are used for training.

The convergence of our iterative parallelized latent rollout algorithm is guaranteed: after finishing the $\ell$-th full iteration, the first $\ell$ blocks of latent tokens become exact without any approximation. Therefore, the proportion of the approximated latent token in the context decreases as the number of iterations increases. As a result, the error of the approximated latent tokens also decreases. Given

---

[2]We can consider the loss on the text tokens for the $\ell + 1$-th stage as the reconstruction loss for the $\ell$-th block of latent tokens.

[3]It requires one additional pass to compute the token probabilities to compute the loss and update the model.

the numerical complexities of the underlying transformers, it is hard to provide theoretical error bounds, and we pick the $L$ empirically. This approach shares root with a concurrent work (Wu et al., 2025) that uses a Jacobi iteration to improve latent CoT during inference time.

**Connection between our method and existing works.** In the extreme cases, when $L = 0$, our rollout method degrades to LightThinker (Zhang et al., 2025), where latent tokens are replaced by fixed textual gist tokens; when $L = B \times m$, our method becomes COCONUT (Hao et al., 2024), which needs to generate all latent tokens autoregressively without any parallelization. The former is less expressive, while the latter is computationally expensive and hard to train. Our method serves as a general framework that unifies existing approaches, offering a balanced trade-off between efficiency and expressiveness, and achieving strong practical training performance.

## 4 EXPERIMENT SETUP

We evaluate our hybrid latent-text reasoning approach on mathematical reasoning benchmarks, comparing against 2 baselines: StreamLLM (Xiao et al., 2023), which modifies attention mask to enable efficient attention computation, and LightThinker (Zhang et al., 2025), which uses textual gist CoT tokens to compress context. In addition, we investigate the trade-offs between inference efficiency and reasoning accuracy when using our proposed iterative rollout algorithm.

**Models and training setup.** We test our models on 7–8B scale, training with Qwen2.5-7B (Team, 2024) and Qwen3-8B (Yang et al., 2025) as base architectures. We train our models using randomly sampled 10k and 50k subsets of OpenThoughts-3 (Guha et al., 2025), which is a collection of high-quality long CoT traces for math and coding questions. We implement our method using a modified version of the llama-factory library (Zheng et al., 2024), utilizing DeepSpeed Zero-3 with offloading (Rasley et al., 2020) to train our models on A100 GPUs. For training, we adopt the same hyperparameters (e.g., learning rate, number of epochs) as Guha et al. (2025) for the 2 scales of the training datasets, respectively. During training, we use a maximum sequence length of 21,000 tokens.

**Datasets and evaluation.** We evaluate models on challenging mathematical reasoning benchmarks including competition math datasets such as AMC'23, AIME'24, AIME'25, the 500-problem subset of MATH (Hendrycks et al., 2021) used by Lightman et al. (2023), and GPQA (Rein et al., 2023). We adopt the same evaluation setup as Muennighoff et al. (2025), using a temperature 0.7, top-p sampling with $p = 1.0$, and a maximum generation length of 32,768 tokens. For AIME'24, Muennighoff et al. (2025) also creates an additional version with figures converted into in-context vector graphics: we report the average performance over the two versions. Given their relatively small dataset sizes, we run evaluations for AMC'23, AIME'24, AIME'25, and GPQA three times using different random seeds, while MATH with more evaluation samples is evaluated once. All runs use bfloat16 precision.[4]

## 5 MAIN RESULTS

**Hybrid CoT achieves 94% performance as text-only CoT with 1.97x less compute.** In Table 1, we compare the performance of our hybrid latent-text reasoning approach against both text-only reasoning baselines and existing efficiency-focused methods across challenging mathematical and scientific benchmarks. Our results demonstrate that hybrid CoT can almost match standard text CoT while providing significant efficiency gains. On Qwen3-8B, our method achieves a 70.96% averaged score across all benchmarks, matching 94.19% of text CoT (75.34%) with 1.97x less compute. On Qwen2.5-7B, our method achieves a 43.84% averaged score across all benchmarks, matching 87.45% of text CoT (50.13%) with 1.59x less compute. This demonstrates that HybridCoT can serve as an efficient counterpart in efficiency-driven applications.

---

[4]While the choice of numerical precision does not significantly impact the baselines, our method performs best with bfloat16, likely because the latent representations were trained in this precision.

Table 1: Model performance on math reasoning benchmarks. "Gen. Len." denotes generation length in tokens.

| | | AIME'24 | AIME'25 | AMC'23 | MC Avg.* | MATH | GPQA | All Avg. | Gen. Len. | # Latent | Comp. ($\times 10^8$) |
|---|---|---|---|---|---|---|---|---|---|---|---|
| | Text CoT | 33.89 | 23.33 | 65.00 | 39.03 | 86.00 | 42.42 | 50.13 | 16.4k | – | 2.16 |
| | | | | | StreamLLM (Xiao et al., 2023) | | | | | | |
| | Sink=4, Window=1024 | 0.00 | 2.22 | 20.83 | 5.76 | 55.00 | 28.79 | 21.37 | 13.7k | – | 0.14 |
| | Sink=Input, Window=1024 | 10.56 | 16.67 | 45.00 | 20.69 | 80.40 | 36.87 | 37.90 | 20.9k | – | 0.24 |
| Qwen2.5-7B | | | | | LightThinker (Zhang et al., 2025) | | | | | | |
| | $m = 9$, block=paragraph | 13.89 | 23.33 | 56.67 | 26.94 | 76.00 | 37.21 | 41.42 | 22.3k | 6.2k | 0.84 |
| | $m = 3$, block=sentence | 10.00 | 11.11 | 44.17 | 18.82 | 74.20 | 34.51 | 34.80 | 22.2k | 2.4k | 0.39 |
| | | | | | **Hybrid CoT (Ours)** | | | | | | |
| | $m = 9$, block=paragraph | 16.67 | 18.89 | 56.67 | 27.22 | 81.00 | 43.27 | 43.30 | 22.0k | 4.8k | 1.57 |
| | $m = 3$, block=sentence | 21.11 | 17.78 | 57.50 | 29.38 | 83.40 | 39.39 | 43.84 | 21.1k | 2.1k | 1.36 |
| | Text CoT | 70.00 | 66.67 | 90.00 | 74.17 | 96.00 | 54.04 | 75.34 | 12.9k | – | 1.42 |
| | | | | | StreamLLM (Xiao et al., 2023) | | | | | | |
| | Sink=4, Window=1024 | 6.67 | 7.78 | 22.50 | 10.90 | 59.00 | 29.12 | 25.01 | 15.2k | – | 0.15 |
| | Sink=Input, Window=1024 | 16.67 | 16.67 | 65.00 | 28.75 | 84.00 | 38.89 | 44.24 | 17.5k | – | 0.20 |
| Qwen3-8B | | | | | LightThinker (Zhang et al., 2025) | | | | | | |
| | $m = 9$, block=paragraph | 46.11 | 36.67 | 80.00 | 52.22 | 91.40 | 52.02 | 61.24 | 20.4k | 4.7k | 0.63 |
| | $m = 3$, block=sentence | 28.89 | 23.33 | 69.17 | 37.57 | 86.60 | 52.19 | 52.04 | 22.8k | 2.6k | 0.40 |
| | | | | | **Hybrid CoT (Ours)** | | | | | | |
| | $m = 9$, block=paragraph | 62.22 | 47.78 | 85.83 | 65.58 | 95.20 | 54.71 | 69.15 | 15.5k | 3.5k | 0.90 |
| | $m = 3$, block=sentence | 61.67 | 54.44 | 88.33 | 66.53 | 94.80 | 55.56 | 70.96 | 14.4k | 1.4k | 0.72 |

\* Given the small sizes of AIME and AMC datasets, we include an average of these three datasets.

**Hybrid CoT significantly outperforms baselines.** When comparing with baselines, HybridCoT significantly outperforms LightThinker, a state-of-the-art latent reasoning method, by 1.36x and 1.26x on Qwen3-8B and Qwen2.5-7B, respectively. In addition, our method provides superior accuracy—70.96% on Qwen3-8B and 43.84% on Qwen2.5-7B—compared with training-free context compression methods like StreamLLM—44.24% on Qwen3-8B and 34.80% on Qwen2.5-7B—that suffer from information loss during compression in the text space.

**HybridCoT works well on complex tasks that requires longer reasoning traces.** Notably, our approach shows particularly strong performance on competition mathematics problems that require long CoT, achieving a 66.53% average on AIME and AMC benchmarks compared to 28.75% for StreamLLM and 37.57% for LightThinker on Qwen3-8B. In addition, HybridCoT achieves an average of 29.38% on long-CoT benchmarks, in comparison with 18.82% for LightThinker and 20.69% for StreamLLM on Qwen2.5-7B.

These results validate that our hybrid approach successfully combines the precision of textual reasoning with the efficiency of latent representations, enabling models to maintain high reasoning performance while reducing computational overhead.

## 6 ABLATION STUDIES

**Determining the best $L$** A key hyperparameter in our iterative rollout algorithm is the number of iterations $L$ used during training. As shown in Fig. 5, we analyze the convergence behavior by measuring the L2 norm of the embedding difference between the approximated latent tokens and the ground truth latent tokens obtained through full autoregressive rollout. The plot demonstrates that the approximation error decreases rapidly in the first few iterations, with the most significant improvement occurring between iterations 0 and 2. Beyond iteration 2, the convergence rate slows substantially, yielding diminishing returns in approximation quality. Based on this analysis, we set $L = 2$ for all our experiments, as it provides the most effective trade-off between approximation accuracy and computational efficiency. This choice allows us to capture the majority of the convergence benefits while maintaining reasonable training costs.

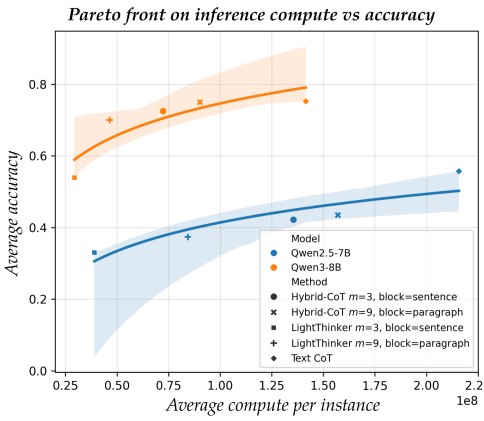

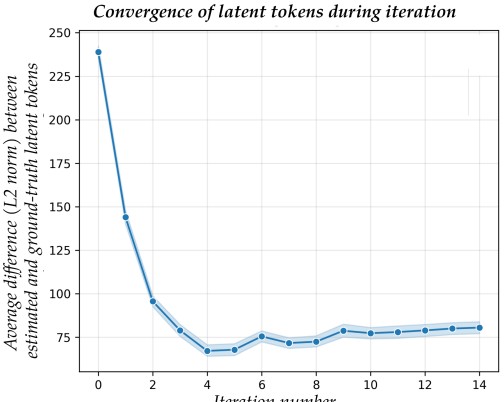

Fig 4: Compared to the baselines, Hybrid-CoT finds a balance between inference compute and model accuracy.

Fig 5: The latent tokens approximation quickly converges after a few iterations in our iterative algorithm.

**Picking $m$ and the block split** In Table 2, we ablate two key design choices: the number of latent tokens per block ($m$) and the granularity of reasoning block splits (sentence vs. paragraph level). We observe that performance generally improves as $m$ increases from 1 to 9 across both splitting strategies, with diminishing returns at higher values. Notably, sentence-level splitting consistently outperforms paragraph-level splitting across all block sizes, achieving a micro average of 43.58% compared to 40.78% at $m = 9$. However, sentence-level splitting introduces computational overhead due to more frequent transitions between text and latent reasoning modes, leading to training slowdowns. Considering this efficiency trade-off, we adopt paragraph-level splitting with $m = 9$ for our main experiments, as it provides competitive performance (within 3% of the best configuration) while maintaining better computational efficiency.

Table 2: We ablate the best configuration of $m$ as well as the how to create the reaosoning blocks (in terms of sentences or paragraphs) for our proposed method.

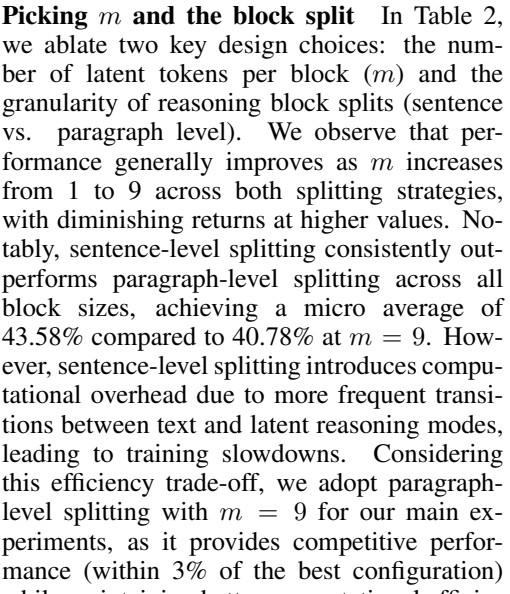

|  | $m$ | AIME'24 | MATH[1] | GPQA | Micro Avg. |
|---|---|---|---|---|---|
| **Sentence** | 1 | 4.44 | 54.33 | 30.64 | 32.87 |
|  | 3 | 3.89 | 69.67 | 35.35 | 39.66 |
|  | 5 | 10.00 | 75.67 | 33.50 | 41.34 |
|  | 7 | 10.00 | 76.00 | 36.03 | 42.83 |
|  | 9 | 13.33 | 80.00 | 34.34 | 43.58 |
| **Paragraph** | 1 | 1.11 | 54.00 | 27.61 | 30.54 |
|  | 3 | 6.11 | 66.00 | 29.29 | 35.66 |
|  | 5 | 6.67 | 70.67 | 32.15 | 38.64 |
|  | 7 | 8.89 | 72.33 | 32.32 | 39.57 |
|  | 9 | 8.89 | 75.33 | 33.00 | 40.78 |

[1] We use a 100 random subset of MATH.

# 7 CONCLUSION

We present HybridCoT, a method that trains a language model to interleave textual and latent CoT steps for reasoning for math problems. The generated text tokens scaffolds the latent tokens, and they are removed from the context to speed up inference. Certain text tokens like math symbols and numbers are retained in the context, which we find to be crucial for math reasoning tasks. To make training of long latent CoT possible, we introduce an iterative parallelized latent rollout algorithm that has a constant complexity irrespective of the reasoning length. Empirical results show that our method achieves 70.96% average performance on Qwen3-8B, matching 94% of text CoT performance while requiring $1.97\times$ less compute. Our method outperforms existing efficient reasoning approaches by $1.36\times$ on average and shows particularly strong results on complex competition mathematics problems. These results demonstrate that hybrid reasoning successfully combines the precision of textual reasoning with the efficiency of latent representations.

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

## A   ANNOTATION MATH SYMBOLS IN THE TRAINING DATA

> You will be given text containing mathematical reasoning. Your task is to identify and wrap all mathematical expressions, variables, and formulas with `<math>` tags.
> Rules:
> - Tag complete mathematical expressions (e.g., equations, inequalities, formulas)
> - Tag individual variables when they appear in isolation
> - Tag numerical computations and their results
> - Do NOT tag mathematical terms written in words (e.g., "parabola", "derivative")
> - Preserve the exact spacing and formatting within tags
> - When multiple mathematical expressions are connected by "and", "or", or commas within the same logical unit, tag them together as one to minimize flow disruption
> - Please do NOT label spans like "Answer:", "Solution:", "Explanation:", "Proof:", "Conclusion:", "Final Answer:", "Final Solution:", "Final Explanation:", "Final Proof:", "Final Conclusion:"
> - Do NOT modify the original text inside tags - copy it exactly as is, without adding LaTeX commands
> - If there are blocks of code, please do NOT generate any tags for them
>
> Examples:
>
> Example 1:
> Input: Given that x + 2y = 10 and x - y = 1, we can solve for x and y. First, from the second equation, x = y + 1.
> Output: Given that `<math>`x + 2y = 10 and x - y = 1`</math>`, we can solve for `<math>`x and y`</math>`. First, from the second equation, `<math>`x = y + 1`</math>`.
>
> Example 2:
> Input: The quadratic formula states that $x = (-b \pm \sqrt{b^2 - 4ac})/2a$. When a = 1, b = -5, and c = 6, we get x = 2 or x = 3.
> Output: The quadratic formula states that `<math>`$x = (-b\pm\sqrt{b^2 - 4ac})/2a$`</math>`. When `<math>`a = 1, b = -5, and c = 6`</math>`, we get `<math>`x = 2 or x = 3`</math>`.
>
> Example 3:
> Input: The constraints are $0 \le x \le 10$, $y \ge 0$, and $x + y \le 15$. The objective function is $z = 3x + 2y$.
> Output: The constraints are `<math>`$0 \le x \le 10, y \ge 0$, and $x + y \le 15$`</math>`. The objective function is `<math>`$z = 3x + 2y$`</math>`.
>
> Example 4:
> Input: Set $A = \{1,2,3\}$ and $B = \{2,3,4\}$, so $A \cap B = \{2,3\}$ and $A \cup B = \{1,2,3,4\}$.
> Output: Set `<math>`$A = \{1,2,3\}$ and $B = \{2,3,4\}$`</math>`, so `<math>`$A \cap B = \{2,3\}$ and $A \cup B = \{1,2,3,4\}$`</math>`.

## B   THEORETICAL ANALYSIS ON CONVERGENCE

**Notations.** Following the notations in Section 3.2, we only consider the latent tokens after the complete $m$ forward passes at the end of each iteration (i.e., $t = m$ in Equation (2)). For simplicity, we denote the latent tokens in the $b$-th block as $Z^{(\ell)}[b]$, $\forall b \in [1, B]$, the aggregation of the first $b$ blocks as $Z^{(\ell)}[1:b] := (Z^{(\ell)}[1], \dots, Z^{(\ell)}[b])$, and the complete $B$ blocks as $Z^{(\ell)} := Z^{(\ell)}[1:B]$. We denote the operator $T : Z \rightarrow Z$ as one full iteration (consist of $m$ updates) of the iterative refinement scheme, i.e., $Z^{(\ell+1)} = T(Z^{(\ell)})$. We denote the operator $F : Z \rightarrow Z$ as the exact block-wise causal latent operator that generate the ground truth latent $Z^*$. Thus, by definition, $Z^*$ is a fixed point for $F$, i.e., $Z^* = F(Z^*)$. We further define the error at iteration $l$ as $e^{(\ell)}[b] := \|Z^{(\ell)}[b] - Z^*[b]\|$ and $e^{(\ell)} := \|Z^{(\ell)} - Z^*\|$.

Both operators $T$ and $F$ have a causal structure where each token depends on only the tokens before it. They also share the same neural network. Therefore, $T$ and $F$ are consistent on exact past, and the operator $T$ generates exact outputs when the iteration number is larger than the number of blocks.

**Proposition 1** (Front Exactness). *For all $\ell \in [1, B]$ and all $b \leq \ell$, $e^{(\ell)}[b] = 0$.*

*Proof.* We prove by induction on $\ell$. At the first iteration, the calculation of the first block does not depend on any past blocks and $e^{(1)}[1] = 0$ holds.

At iteration $\ell$, if the first $\ell$ blocks are exact at input, i.e., $Z^{(\ell)}[1 : \ell] = Z^*[1 : \ell]$, then based on consistency between $T$ and $F$, we have

$$Z^{(\ell+1)}[b] = T(Z^{(\ell)})[b] = Z^{(\ell)}[b] = Z^*[b], \ \forall\, b \leq \ell,$$

and

$$Z^{(\ell+1)}[\ell + 1] = T(Z^{(\ell)})[\ell + 1] = T(Z^*)[\ell + 1] = F(Z^*)[\ell + 1] = Z^*[\ell + 1].$$

Thus, the statement holds for iteration $\ell + 1$. By induction, it holds for all $\ell \in [1, B]$. $\qquad\square$

Further, we examine the convergence property for the cases when the iteration number is small. We build our analysis on the following two assumptions:

**Assumption 1** (Lipschitz Property of $F$). *There exists $\rho \in [0, 1)$ such that for any latent $Z$ and $Z'$, $\|F(Z) - F(Z')\| \leq \rho\|Z - Z'\|$.*

**Assumption 2** (Tapered Approximation Gap). *There exists a non-increasing sequence $\{C_\ell\}_{\ell \geq 1}$ with $C_\ell \downarrow 0$ such that, at iteration $\ell$, $\|T(Z^{(\ell)}) - F(Z^{(\ell)})\| \leq C_\ell\|Z^{(\ell)} - Z^*\|$.*

Assumption 1 indicates that $F$ is a contraction so that repeated application of $F$ would drive the initial state towards the unique fixed point $Z^*$. Assumption 2 follows the intuition from Proposition 1 that $Z^{(\ell)}$ contains more exact components as the iteration number $\ell$ increases. Thus, the approximation $T$ is closer to the exact operator $F$ as as more blocks become exact, leading to $C_\ell$ converging to zero. Based on these assumptions, we can derive the eventual linear convergence for the approximation error.

**Proposition 2** (Eventual Linear Convergence). *Under Assumption 1 and Assumption 2, there exists $L_0 \in \mathbb{N}$ such that $\theta := \rho + \sup_{\ell \geq L_0} C_\ell < 1$. Consequently, for all $\ell \geq L_0$,*

$$e^{(\ell+1)} \leq \theta\, e^{(\ell)},$$

*and hence for all $k \geq 0$,*

$$e^{(L_0+k)} \leq \theta^k\, e^{(L_0)}.$$

*Proof.* Using $Z^* = F(Z^*)$ and the triangle inequality, we have

$$
\begin{aligned}
e^{(\ell+1)} &= \|Z^{(l+1)} - Z^*\| \\
&= \|T(Z^{(\ell)}) - F(Z^*)\| \\
&\leq \|F(Z^{(\ell)}) - F(Z^*)\| + \|T(Z^{(\ell)}) - F(Z^{(\ell)})\| \\
&\leq \rho\,\|Z^{(\ell)} - Z^*\| + C_\ell\,\|Z^{(\ell)} - Z^*\| \\
&= (\rho + C_\ell)\, e^{(\ell)},
\end{aligned}
$$

where we used Assumption 1 and Assumption 2 in the second inequality. Since $C_\ell$ is non-increasing and $C_\ell \to 0$, there exists $L_0$ with $\theta := \rho + \sup_{\ell \geq L_0} C_\ell < 1$. For all $\ell \geq L_0$ we then have $\|e^{(\ell+1)}\| \leq \theta\|e^{(\ell)}\|$, which yields the geometric decay

$$e^{(L_0+k)} \leq \theta^k\, e^{(L_0)}, \quad \forall\, k \geq 0.$$

$\qquad\square$

If we further add a stronger assumption to the decay rate of $C_\ell$ in Assumption 2, i.e., an exponential decay rate as in Assumption 3, there is a sharper convergence rate for the approximation error $e^{(\ell)}$ as shown in Proposition 3.

**Assumption 3** (Exponential Taper). *There exists $\beta \in [0, 1)$ such that, at iteration $\ell$, $\|T(Z^{(l)}) - F(Z^{(l)})\| \leq \beta^\ell \|Z^{(l)} - Z^*\|$.*

**Proposition 3** (Sharper Convergence under Exponential Taper). *Under Assumption 1 and Assumption 3, we have for all $\ell$,*

$$e^{(\ell+1)} \leq (\rho + \beta^\ell)\, e^{(\ell)} \quad and \quad e^{(\ell)} \leq \left( \prod_{j=1}^{\ell-1} (\rho + \beta^j) \right) e^{(1)} \leq \rho^{\ell-1} \exp\left( \frac{\beta}{\rho\,(1-\beta)} \right) e^{(1)}.$$

*Proof.* Following the same derivation as in Proposition 2, we have

$$e^{(\ell+1)} \leq (\rho + \beta^\ell)\, e^{(\ell)}.$$

Iterating the inequalities over $\ell$ yields

$$e^{(\ell)} \leq \left( \prod_{j=1}^{\ell-1} (\rho + \beta^j) \right) e^{(1)}.$$

To obtain the explicit upper bound, note that for $j \geq 0$,

$$\rho + \beta^j = \rho \left( 1 + \frac{\beta^j}{\rho} \right) \implies \prod_{j=1}^{\ell-1} (\rho + \beta^j) = \rho^{\ell-1} \prod_{j=1}^{\ell-1} \left( 1 + \frac{\beta^j}{\rho} \right).$$

Using $\log(1 + x) \leq x$ and summing the geometric series,

$$\prod_{j=1}^{\ell-1} \left( 1 + \frac{\beta^j}{\rho} \right) = \exp\left( \sum_{j=1}^{\ell-1} \log\left( 1 + \frac{\beta^j}{\rho} \right) \right) \leq \exp\left( \sum_{j=1}^{\ell-1} \frac{\beta^j}{\rho} \right) \leq \exp\left( \frac{1}{\rho} \cdot \frac{\beta}{1-\beta} \right).$$

This leads to the stated bound. The product decays geometrically with leading factor $\rho^{\ell-1}$ up to a multiplicative constant.

$\square$