# OpenReview forum: "HybridCoT: Interleaving Latent and Text Chain-of-Thought for Efficient Reasoning"
_ICLR.cc/2026/Conference — Submitted to ICLR 2026_

### Official Review · Reviewer_Q1Af · 2025-10-15

**Soundness:** 2
**Presentation:** 1
**Contribution:** 2
**Rating:** 2
**Confidence:** 3

**Summary:**

This paper introduced Hybrid-CoT a method for using both soft (latent) and hard tokens at test time. The key part to this paper is that the full context is not replaced with soft tokens, some hard tokens are kept in the context.

During inference the model decodes any number of hard tokens followed by a \<latent\> token, which switches the model into producing m (fixed hyperparam) number of soft tokens.
Each token has the choice to attend to either: previous math tokens (marked with \<math\> tokens prior by the model); all previous latent tokens; or all tokens in the current block. This means that the attention is very sparse, hence fast.

The model is trained by having a stronger model annotate \<math\> tags into the training data and \<latent\> tokens are added at the end of paragraphs or sentences. So the trained model can effectively use both \<math\> and \<latent\> tags.
To avoid most of the increased cost of training a COCONUT style objective the authors use an approximation.
This approximation allows all latent tokens to be rolled out in a smaller number of iterations by using older intermediate latent token values for future latent tokens instead of always using the newest; meaning multiple latent tokens can be rolled our concurrently, the level of approximation is controlled by a hyperparameter.
Overall the authors achieve noticeably higher results than the baselines and are more efficient over many difficult benchmarks.

**Strengths:**

- Compares on two large models, although they are both Qwen based.
- Is more efficient and achieves higher accuracy on benchmarks.

**Weaknesses:**

- Number of latent/soft tokens decoded is a hyper parameter and therefore fixed.
- Method and writing feels overly complicated, simple explanations would benefit the paper a lot.
    - Section 3.2 is not well explained, please make this much clearer. The convergence guarantee here is hand wavy, please make it rigorous or remove it. I would say this is the highest priority during rebuttal.
- Very math benchmark focused.
- Lacking baselines: around mixing soft and hard tokens e.g. https://arxiv.org/pdf/2505.18962 (May 2025), https://arxiv.org/pdf/2502.21074 (Feb 2025)
- Minor: please bold the highest numbers in Table 1 to make it more readable.

I think the main issues with this paper are lack of reasoning as to why hyperparameters are chosen and presentation of the new method.

**Questions:**

1. Why not use the model which annotates \<math\> tokens to also add \<latent\> tokens, instead of using sentences and paragraphs?
2. The training data is obtained by using a strong model to annotate data. Is this unfairly distilling extra information from the stronger model into your method?
3. How does this method impact memory usage versus the baselines?
4. Figure 5 decreases even more for L=3 or 4, why choose L=2 over these?
5. Table 2 only considers m up to 9, but the accuracy is still increasing, why stop at 9?
6. Does this method extend outside of the Qwen family?

---

> ### Author Response · Authors · 2025-12-04
> **Response to Reviewer Q1Af**
>
> Thank you for your review.
>
> **[Weakness 1: our method uses fixed latent tokens]**
>
> There might be some confusion in this question (given the reviewer's follow-up question).
> - First, our method has **a dynamic total number of tokens** per generation. This is because the trained model can dynamically decide to switch to latent reasoning when it needs to, and thus the model can dynamically decide whether to have more rounds of latent reasoning given the difficulty of the problem.
> - Second, we find it helpful that we can control the model to generate **a fixed number of latent tokens per block**. It is actually beneficial because it makes it easier to control the generation process and keeps training stable. Compared to having a model generate dynamic numbers of latent tokens per block, the fixed latent token per block gives us a knob to change the trade-off between efficiency and accuracy.
>
> **[Weakness 2: complicated Section 3.2]** We think the reviewer is primarily concerned with the writing of the method section 3.2, which we’ve updated writing (see general response 4).
>
> **[Weakness 3: convergence guarantee is not rigorous]**
> We’ve included a proof on this in appendix B. The high-level intuition is that, as the iteration number increases, the number of approximated tokens decreases, and the errors on the approximated tokens also decrease.
>
> **[Weakness 4: Very math benchmark focused]**
> First, it's common to use models' performance on various math benchmarks as important indicator of their reasoning capabilities. We’ve conducted extensive experiments on four math datasets (AIME 24, 25, AMC 23, and MATH), which is aligned with previous work like [open-thoughts](https://arxiv.org/pdf/2506.04178) and [s1: Simple Test Time Scaling](https://arxiv.org/pdf/2501.19393).
> Moreover, we also tested on GPQA, which targets PhD-level science questions beyond math. Our method also achieved strong results on that dataset: in fact, the best performing model using our method is better than the baseline method (e.g., 54.71 vs 54.04) while using a fraction of the compute (0.90 vs 1.42). This shows our method demonstrates strong capability outside math domains.
>
> **[Weakness 5: Lacking baselines]**
>
> We thank the reviewers for mentioning more baselines. For the System-1.5 Reasoning paper by Wang et al., there is no open-sourced code available. As the method is relatively complicated with a lot of algorithmic details, we do not think we can reproduce the work in the given time. We will include CODI results in the final draft of the paper.
>
> **[Weakness 6: Bold numbers in the table]** Yes, we will update it in the final draft.

---

> > ### Author Response · Authors · 2025-12-04
> > **Response to Reviewer Q1Af**
> >
> > **[Question 1: Why not use the model which annotates `\<math\>` tokens to also add `\<latent\>` tokens, instead of using sentences and paragraphs?]**
> > Actually this is what we do during inference. During the training stage, the language model learned when to insert the `\<latent\>` token given the training data, which we preprocess to insert the `\<latent\>` tokens  at sentence or paragraph breaks. During inference, the model automatically generates the `\<latent\>`, which triggers the switch to the latent decoding mode.
> >
> > **[Question 2: The training data is obtained by using a strong model to annotate data. Is this unfairly distilling extra information from the stronger model into your method?]**
> > We respectfully disagree with your statement.
> > - We made sure that the model **only inserted math tokens** in the data generation process. As we stated in appendix A, in fact, we tried very hard to avoid any deviation from the original text: after the prompting, if there’s a clear difference in the generated text (after removing math tags), we will discard the model generation and use the original text. Our goal is to make sure that the data is consistent.
> > - Further, to ablate the potential concerns that inserted `\<math\>` can improve the training, we trained a text CoT model on the same data with `\<math\>` tokens inserted. We do not observe visible differences across the benchmarks. We will include the results in the final draft.
> >
> > Finally, in case there are further confusions, we want to clarify that in our experiments, we train the text CoT, LightThinker, and our method on the same OpenThoughts dataset. The streamLLM method is applied to our trained text CoT models.
> >
> > **[Question 3: How does this method impact memory usage versus the baselines?]**
> > Our method reduces memory footprint compared to text CoT by removing previously generated text from the KV cache, which lowers memory usage during inference. We will add a detailed analysis of memory footprint in the final draft.
> >
> > **[Question 4: Figure 5 decreases even more for L=3 or 4, why choose L=2 over these?]**
> > $L=2$ is chosen empirically based on the trade-off between efficiency and performance. While larger iteration numbers yield smaller approximation errors, they add training overhead without significant performance gains. We also hypothesize that the small errors from lower iteration counts act as regularization during training, making the model more robust during inference.
> >
> > **[Question 5: Table 2 only considers m up to 9, but the accuracy is still increasing, why stop at 9?]**
> > In early experiments, we found that further increasing $m$ reduces efficiency without yielding substantial performance improvements.
> >
> > **[Question 6: Does this method extend outside of the Qwen family?]**
> > Yes, our method should extend to other model families. The core components of our method rely only on standard transformer operations that are common across all modern LLMs. There are no Qwen-specific architectural dependencies. We validated our approach on two Qwen models (Qwen3-7B and Qwen3-8B), and observed consistent improvements across both. This provides preliminary evidence of generalizability within varied model configurations.

---

### Official Review · Reviewer_cvby · 2025-10-30

**Soundness:** 4
**Presentation:** 3
**Contribution:** 4
**Rating:** 8
**Confidence:** 3

**Summary:**

The paper proposes a new approach, HybridCoT, whose aim is to reduce the length of the reasoning chains via the use of "Thinking Tokens". These tokens are interleaved with some regular tokens (i.e. explicit math tokens), to avoid over-compressing the reasoning chains.

The method works as follows. Given a reasoning chain made up of multiple blocks (e.g. sentences or paragraphs), each block is interleaved with $m$ latent tokens. This structure enables the latent tokens to capture information specific to the given reasoning block (a la gist token), and be the main conduit of that information to the next block. That said, the authors allow for some relaxation of this, where some special tokens (math tokens), can also be attended to, in order to avoid over-compressing the chains.

Importantly, the latent tokens are continuously generated, i.e. are not limited to a fixed set of augmented tokens. A learned linear transformation is applied on top of the penultimate hidden representation before feeding it back to the Transformer.

The training procedure wraps the latent token insertion with special <latent> tokens. Crucially, the attention mask is modified so that a given token can only  attend to 1) math tokens from previous blocks, 2) the latent tokens from any previous step and 3) any text token in the current block. This forces the latent tokens to capture the block-level information. By predicting the text tokens at every step, this design allows for each latent token to receive an intermediate signal.

The authors then propose a way to alleviate the slow training procedure, which whould require as many forward pass as `number_of_blocks` x `number_of_latent_tokens_per_block`. They show than one can reduce this cost to `number_of_blocks`, which can further be reduced if allowing for an approximation error.

Experiments are conducted on a wide variety of mathematical reasoning benchmarks using the Qwen family of models. The OpenThoughs dataset is used for training the approach. Experiments are convincing, showing that the proposed approach navigates well the compute / performance tradeoff. Finally, the authors show via ablation studies that the training approximation of the iterative parallelized rollout indeed converges within a reasonable margin of error.

**Strengths:**

1. The approach is nicely designed. The interleaving of the latent tokens to capture step-wise information, the resulting finegrained training signal from the text token prediction, to the iterative parallelized latent rollout, the overall approach is well executed.
2. The paper is well presented, the experiments are well targeted, and the ablations are properly built.

**Weaknesses:**

1. I am somewhat hesitant about the tailoring of the approach to math specific problems. How would you identify the special tokens which should not be compressed in settings outside of math ?

**Questions:**

1. How exactly are the gist tokens (used  in section 3.2) initialized ? Is this initialization learned ?
2. Why isn't the error in Figure 5 going to zero ? Given that the approximation is exact after $l$ steps for the first $l$ blocks, shouldn't it converge to zero ?

---

> ### Author Response · Authors · 2025-12-04
> **Response to Reviewer cvby**
>
> Thank you for your review and detailed summary of our method!
>
> **[Weakness 1: Retaining only math spans is too specific to math problems]**
> We agree that it is a limitation that the current method only learns to keep math spans. However, as we mentioned in footnote 1 (line 215), we can test it in a controlled setting and it helps verify our assumptions.
> To make it more generalized, as we mentioned in our submission (line 176), we can use, for example, a reinforcement learning based approach that learns which token spans are most important to keep in the context. By optimizing the compression ratio as well as the final accuracy, one can learn a better function to determine what spans to keep.
>
> **[Question 1: Initialization of the gist tokens]**
> As we mentioned in General Response 3: the initial version of the latent tokens (z at step 0, or the gist tokens) is computed as the average embedding of all tokens. In our Qwen-based experiments, since there are reserved unused tokens in the vocabulary, we simply use the embedding from token ID [151669, ..., 151677].
>
>
> > Why isn't the error in Figure 5 going to zero ? Given that the approximation is exact after  steps for the first  blocks, shouldn't it converge to zero ?
>
> **[Question 2: Convergence of the trend in Figure 5]** We find that this issue is caused by numerical inaccuracies in A100 GPUs (due to the accumulation of floating point errors in the latent rollout processes). The issue disappeared when we switched to H100 GPUs.

---

### Official Review · Reviewer_Q6MN · 2025-10-30

**Soundness:** 3
**Presentation:** 2
**Contribution:** 2
**Rating:** 4
**Confidence:** 3

**Summary:**

This work presents a method for combining textual and latent reasoning methods in LLMs to increase both training efficiency and test accuracy. Existing interleaved latent and textual reasoning token approaches tend to have to roll out all intermediate latents autoregressively within a single sequence before the actual training update can be computed which greatly increases training costs over standard text only training.  However their approach trains for this inference behavior using a more efficient parallelized iterative strategy that caps costs at a lower number of forward evaluations by "relaxing the causal dependency" between blocks containing latent tokens. They present promising results showing near parity in terms of accuracy with text based CoTs and report efficiency improvements over their selected baselines.

**Strengths:**

1. The method is presented as a generalization of prior work on incorporating latent reasoning tokens (Lightthinker, COCONUT) suggesting the pareto-optimality of their solution (though this is hard to evaluate, see concerns)
2. They perform experiments on full size models like Qwen3-8B rather than 1B or less
3. Their more efficient method recovers most of the performance of the standard text only CoT across the series of benchmarks considered for both models.

**Weaknesses:**

The points and questions below limit the ability of the reader to asses the novelty of the technique versus other methods and its reported efficiency edge. The current rating for the paper is primarily based on these issues and so their adequate resolution could improve the reviewer's assessment.

### Efficiency results are not communicated clearly

### 1.

"1.97× less inference compute" is a confusing way to report improvement, please use a "achieve a 30% reduction" or "uses 50% of" to describe how the method is more compute efficient, unless the metric is tokens per second, or something where more is better, and in that case, say "1.97x tokens per second" or something explicitly.

### 2.

What is "Comp. (x10^8) in Table 1? In general it is not clear how computational cost/efficiency is computed and then compared for text CoT versus, StreamLLM versus Lightthinker, versus Hybrid CoT

### Description of latent token implementation is unclear

### 3.

It is unclear how the L163 comment is supposed to differentiate this approach from Lightthinker. Later in S3.2 the "gisting" vocabulary is introduced, so, does this method also use a fixed vocabulary of latent special tokens g_1, g_2, etc added to the embedding matrix, similar the the fixed set of caching tokens? Is the maximum number of latents possible per block then limited by the chosen value of m before training starts? L274 attempts to help here but does not provide enough clarity.

I think that one issue is the repeated use of the word "fixed" throughout the draft which doesn't have a clearly defined meaning in this context. Can the authors precisely explain what is "fixed" about Lightthinker, and how this matches the L=0 case of their algorithm? In my mental model, while a special token in the vocabulary and its embedding vector stored in the embedding matrix are in one sense "fixed" at test time, when they are actually passed in as input to the first layer of the model, as the forward pass executes, at the spec token's position a column of activations is created in the model (KV states) that are _not fixed_ at all and rather are dynamic data dependent representations of previous tokens in the context, so, calling this type of operation the used of "fixed" tokens is odd.

### 4.

More generally, S3.2 is quite hard to understand, though I see that it is perhaps the most complex part of the method so maybe this is expected. The attention mask provided in Fig 2 is helpful I think, but it's not clearly connected to Fig 3. In the iterative parallelized latent rollout diagram in S3.2, are the sequences X'[1] and X'[2] in Fig 3 intended to correspond to x0,x1,x2, and x4,x5 in Fig 2? I noted the phrase "relax the causal dependencies across reasoning blocks" and that the cost is meant to be iteration depth l by the num masks per block m but really I struggled to get anything out of Figure 3. What does the t index refer to? positions appear to be indexed as columns in Fig. 3 so i can't tell how "t-th token generation in the l-th iteration" should be interpreted in context.

I dont think this is the case, but to clarify, the accompanying attention mask for iterations l=1, l=2 etc. eg what gets to attend to what while this latent iteration is being performed is the same at every iteration right? I suspect that this mask is constant as a function of l, but varies for each complete training step based on where the block boundaries, text tokens, and latent tokens happen to fall. Does relaxing the causal dependency mean that in the missing attention mask diagram corresponding to Figure 3, would z2,1 and z2,2 not be able to attend to z1,1 and z1,2 but would be able to attend to X'[1]~=x1,x2,x3 ?

### 5.

Ablation on m and block is not clearly described as the terms sentence level and paragraph level are not made precise nor interpretable. It would be much more clear to represent the choice as the num tokens per block and a ratio of latent tokens to text tokens, then average block size, and latents per block can be reported for each training setting and testing benchmark. The main reason for this request is that for wikipedia or news like content, perhaps, assuming a standard tokenizer, one can intuit the ratios, but for the main experimental settings like AIME and MATH, the notion of sentences and paragraphs aren't as well defined and might be differently distributed.

**Questions:**

### 1.

Does the final model generate reasoning tokens automatically? Or at inference time, based on the block size and latent toks per block params, are the special latent tokens inserted using logic external to the LLM forward itself? It is also unclear whether the training iteration depth l is relevant during test time generation.

---

> ### Author Response · Authors · 2025-12-04
> **Response to Reviewer Q6MN**
>
> Thank you for your reviews and helpful feedback!
>
> **[Weakness 1: Efficiency results are not communicated clearly]** Please refer to General Responses 1 and 2 for detailed explanations.
>
> **[Weakness 2: Description of latent token implementation is unclear]** Please refer to General Response 3, where we clarify the difference between soft tokens and latent tokens and discuss how latent tokens are initialized.
>
> To further clarify, we highlight three key differences between our method and LightThinker:
>
> |                | Training                 | Inference: Generation | Inference: Representation |
> |----------------|--------------------------|----------------------|---------------------------|
> | LightThinker   | Embedding Vector         | Filled               | Fixed Vocabulary          |
> | Our method     | Approximation Algorithm  | Decoded              | $\mathbb{R}^d$            |
>
> > Later in S3.2 the "gisting" vocabulary is introduced, so, does this method also use a fixed vocabulary of latent special tokens g_1, g_2, etc added to the embedding matrix, similar the the fixed set of caching tokens? Is the maximum number of latents possible per block then limited by the chosen value of m before training starts?
>
> This question concerns the difference during inference:
> - Latent tokens do not have a constrained vocabulary. LightThinker uses gist tokens, which are a fixed set of vectors repeated during inference: $[x_{(1, 1)}, \dots, x_{(1, k_1)}, g_1, \dots, g_m, x_{(2, 1)}, \dots, x_{(2, k_2)}, g_1, \dots, g_m, \dots]$. The same set of gist token vectors $(g_1, \dots, g_m)$ is reused across blocks. In our method, the latent tokens are outputs from the language model's previous decoding step and can span $\mathbb{R}^d$: $[x_{(1, 1)}, \dots, x_{(1, k_1)}, z_{(1,1)}, \dots, z_{(1,m)}, x_{(2, 1)}, \dots, x_{(2, k_2)}, z_{(2,1)}, \dots, z_{(2,m)}, \dots]$. Each block has its own unique latent tokens $(z_{(i,1)}, \dots, z_{(i,m)})$.
> - Latent tokens are "decoded" rather than filled in. Since gist tokens are fixed, when the model decides to switch to the "gisting/compression" step in LightThinker, their algorithm simply inserts the same tokens into the context. In our method, the latent tokens are autoregressively decoded.
>
> > Can the authors precisely explain what is "fixed" about Lightthinker, and how this matches the L=0 case of their algorithm?
>
> This question concerns the mechanism during training. Section 3.2 describes an approximate algorithm for rolling out the latent tokens. When L=0, we can simply use the fixed vocabulary (like gist tokens) during training to approximate the latent tokens. However, at inference time, we still "decode" the tokens autoregressively.
>
>
> **[Weakness 4: Ablation on m and block size is not clearly described]** Thank you for the comment here! We will include the average block size and compression ratio in the tables.
>
>
> **[Question 1: How are latent tokens generated?]**
> > Or at inference time, based on the block size and latent tokens per block params, are the special latent tokens inserted using logic external to the LLM forward itself?
>
> As we explained above, in our method, the latent tokens are autoregressively generated rather than inserted (different from LightThinker). The model also decides when to switch to the latent mode itself (as it learned to generate the \<latent\> token given the training data). There is no external logic outside the LLM forward pass.
>
> > It is also unclear whether the training iteration depth l is relevant during test time generation.
>
> No. The iteration depth is only used for the approximation of latent tokens at the training time, and the iteration won’t be used in the inference (as all the latent tokens will be generated one by one).

---

### Official Review · Reviewer_GRT4 · 2025-11-01

**Soundness:** 3
**Presentation:** 3
**Contribution:** 2
**Rating:** 4
**Confidence:** 4

**Summary:**

This paper proposes HybridCoT, a method that allows a language model to interleave textual and latent CoT in the reasoning trace for math problems. For each block in the reasoning trace, LLMs first generate a text segment and then generate a fixed number of latent vectors. Non-critical texts in the previous block will be removed to increase the inference efficiency. The authors also propose a parallelized latent rollout algorithm to vastly reduce the training cost compared to previous work. The performance on several benchmarks shows that HybridCoT achieves similar accuracy to textal CoT but uses fewer compute, achieving a balance between accuracy and efficiency.

**Strengths:**

1. The idea of interleaving textual and latent CoT is novel.
2. The proposed training method is efficient compared to previous work, such as coconut.
3. It strikes a balance between accuracy and efficiency, as demonstrated by experimental results on various benchmarks.

**Weaknesses:**

1. The performance of hybridCoT is slightly worse than Text CoT. So the main advantage of the proposed method should be inference efficiency. The authors measured it using inference compute. I wonder how the compute is calculated. Can the 2x faster compute be translated to, e.g., lower inference latency or using fewer GPUs to support inference? This is currently unclear to me.

2. (minor) In line 127, when discussing the benefit of latent CoT, “It needs fewer decoding steps …”, it might be worth discussing the previous work [1], which theoretically demonstrates the benefit.

**References**:

[1] Zhu, Hanlin, Shibo Hao, Zhiting Hu, Jiantao Jiao, Stuart Russell, and Yuandong Tian. "Reasoning by Superposition: A Theoretical Perspective on Chain of Continuous Thought." arXiv preprint arXiv:2505.12514 (2025).

**Questions:**

1. How is the inference compute measured/calculated? Is it proportional to the inference latency? How is the inference latency of hybridCoT compared to SFT?
2. Why does the 7B model consume more compute than the 8B model? Is it because the output is longer (e.g., more verbose or more inefficient) for the 7B model?
3. In Figure 4, for the orange (8B model) curve, why does the accuracy of Text CoT look lower than hybridCoT?

---

> ### Author Response · Authors · 2025-12-04
> **Response to Reviewer GRT4**
>
> Thank you for your reviews and helpful feedback!
>
> **[Weakness 1: Efficiency and compute calculation]** Please refer to General Responses 1 and 2 for detailed explanations.
>
> **[Weakness 2: Missing reference to prior work on latent CoT benefits]** Thank you for the pointer. We have updated the paper to include this reference in the revised draft.
>
> **[Question 1: Why does 7B consume more compute than 8B?]** As clarified above, `comp.` in the table represents total effective context length during inference. The 7B model generates longer outputs in most cases (see the `Gen. Len.` column in table 1), resulting in a larger value. Additionally, in the newly added FLOPs computation (General Response 1), we compute the actual total inference FLOPs, accounting for both context length and model size.
>
> **[Question 2: Why does Text CoT appear lower than HybridCoT for 8B in Figure 4?]** We identified that this was caused by different averaging methods—the plot used macro averaging while the table used micro averaging. We will correct this discrepancy and update the figure in the revised paper.

---

### Author Response · Authors · 2025-12-04
**Message to AC**

We thank the reviewers for their thoughtful feedback. We are encouraged that CvBy finds our approach "nicely designed" and the experiments "well targeted." We also appreciate GRT4's recognition that "the idea is novel" and "strikes a balance between accuracy and efficiency," and Q1Af and Q6MN find our method to be efficient.

We observe that most of the mentioned weaknesses (and presumably the associated scores) were due to our weaknesses in description, in particular our method description in section 3.2 (mentioned by reviewers Q6MN, Q1Af) and our description of inference efficiency benefits (mentioned by GRT4, Q6MN, Q1Af). We believe that these presentation issues can be largely addressed in the final updated manuscript (and described in more detail) below, and therefore kindly ask the AC to take this into account in making their final decision.

**We've already added proof of the convergence of our approximation algorithm in appendix B, and here is a summary of planned updates in the final manuscript**:

- Writing updates:
    1. Update the method section 3.2 for better presentation of our method
    2. Update details on the initialization of the latent tokens and contrast other methods
    3. Small fixes on the paper text and writeups
      - Bolden the numbers in the table
      - Replot the conference figure using H100
      - Fix the pareto figure b/c of different data aggregation

- Additional experiments/results:
    1. Clarification on the compute and added FLOPs computation
      - Add efficiency analysis of our method, including latency as well as memory footprint
    2. Additional ablations on the impact of the <math> token for the training data
    3. Additional baselines including CODI

---

> ### Author Response · Authors · 2025-12-04
> **[General Response 1] Clarification on how "compute" is calculated**
>
> In Table 1, `comp.` is calculated as the average "total effective context length" for each method. For example, for the default full attention decoding, the total effective context length $L$ is quadratic with respect to the generation length $1+2+\dots+L=0.5L(L+1)$; and for StreamLLM it is linear with respect to the window size $\approx w \times L$. For our method and LightThinker, the trained model dynamically decides when to switch to latent thinking and removes the context—we compute this based on the attention mask used during generation.
>
> We acknowledge that this is an approximate measurement of the actual compute incurred. We will include the total FLOPs during inference in the paper and add a more detailed description of the FLOPs calculation in the final draft.
>
> Further, as reviewers GRT4 and Q6MN mentioned, the "x1.96 faster" phrasing is confusing, and we will update the description in the revised draft.

---

> ### Author Response · Authors · 2025-12-04
> **[General Response 2] Additional efficiency analysis of the proposed method**
>
> We will provide additional details on how our method reduces computational latency in the final draft. Our method reduces inference latency because it uses a shorter effective context length: this not only reduces attention computation but also lowers latency due to the smaller KV cache size (less memory transfer overhead).

---

> ### Author Response · Authors · 2025-12-04
> **[General Response 3] Additional details on latent token implementation**
>
> Following the suggestion by reviewers Q6MN and CvBy, we clarify the difference between text tokens, soft tokens, and latent tokens:
> - **Text tokens** are tokens in the vocabulary of a language model. The model stores a **fixed** embedding vector for each token.
> - **Soft tokens** are special tokens beyond the standard vocabulary. They are similar to text tokens in that the LM stores fixed embedding vectors for them, but they do not have clear lexical meaning, for example “\<gist1\>”, "\<gist2\>", etc. There is a **fixed number** of such tokens: for example, Mu et al. experimented with 2, 5, or 10 gist tokens.
> - **Latent tokens** are arbitrary **vectors** in $\mathbb{R}^n$ (the same representation space as the token embeddings). They are direct outputs from the language model with no constraints on them during inference and can theoretically span the entire $\mathbb{R}^n$.
>
> Regarding the initialization of latent tokens: as mentioned in Section 3.2 (line 252), the iterative algorithm requires an initial value at the first iteration. In general, this can be computed as the average embedding of all tokens. In our Qwen-based experiments, since there are reserved unused tokens in the vocabulary, we simply use the embeddings from token IDs `[151669, ..., 151677]`.

---

> ### Author Response · Authors · 2025-12-04
> **[General Response 4] Improving the presentation of Section 3.2**
>
> We will improve the method section 3.2 for better presentation of the method. This concern was raised by Q6MN and Q1Af, and our primary changes are to clarify that (1) our proposed approximation algorithm is only applied at training time, not inference time, and (2) incorporate the newly added proof and high-level intuition in the explanation of the algorithm.

---

### Meta-Review · Area_Chair_yaiP · 2026-01-06

**Summary:**

The submission "HybridCoT: Interleaving Latent and Text Chain-of-Thought for Efficient Reasoning" describes a training scheme and inference method capable of equipping an existing model with a mixture of latent and standard chain-of-thought reasoning, interpolating between latent reasoning methods like coconut and text compression methods like StreamLLM and LightThinker. The submission argues that this recovers the same performance as standard CoT, but with increased inference performance.

**Reviewer Concerns:**

Reviewers raised several concerns during the rebuttal that I found notable:
* *Concerns regarding the communication of efficiency results*. The submission argues that the main advantage of the proposed method is efficiency, however, 'efficiency' is mainly tracked by 'total effective context length', and an approximate FLOP calculation added during rebuttal that is not being discussed in the body of the paper. Overall, the reader is left uncertain whether the proposed approach is actually more efficient. Reviewers asked for measurements of "lower inference latency" or hoped that this method  would allow the authors to use "fewer GPUs to support inference", but clear answers to these questions were not provided during rebuttal.
* The paper uses a handcrafted rule to decided which tokens to track in latent space and which in discrete space, using a large model to label spans of math. While the authors argue that this could be learned in general and could be applied in other domains, this is not shown. Further, it would be important to separately ablate everything happening with interleaving based on math spans. Are the advantages coming from the math span labeling, i.e. what does performance at L=1, or L=0 look like?
* Lack of baselines. The submission compares to two closely related baselines from the token compression literature, but reviewers are concerned that no latent-reasoning method is compared to (like coconut) and no other interleaved reasoning like " https://arxiv.org/pdf/2505.18962 (May 2025), https://arxiv.org/pdf/2502.21074 (Feb 2025)".
* Finally there were concerns about the amount hyperparameters in the method.

There were also concerns about unclear writing by several reviewers, but reading the rebuttal comments and the updated draft, I consider these concerns to be resolved. One reviewer was concerned about the convergence of parallel trainining scheme discussed in Section 3.2. The authors provided a proof in the rebuttal, but I looked into the proof, and did not think that it was illuminating. Arguably the proof is restating the Banach fixed-point theorem (that contraction implies linear convergence), which is not helpful in understanding why the proposed scheme would result in a contractive operator.

Finally, I have the concern whether L=2 is sufficient to really learn latent reasoning like in coconut, especially when doing parallel inference over a long sequence. In effect, the training signal here is quite shallow, and it seems like this moves the method quite close to lightthinker in effect? Here I really would have liked to see the impact of models trained with different values of L.



Overall, due to the concerns raised during the review I am arguing for the current version of this submission not to be accepted.

**Reviewer Scores:**

GRT4: remain at 4;
Q6MN may have raised to 6;
cvby remain at 8;
Q1Af likely no raise from 2 due to concerns about proof

---

### Decision · Program_Chairs · 2026-01-26

Reject